# Spectroscopic Investigation of DCCH and FTSC as a potential pair for Förster Resonance Energy Transfer in different solvents

**Georg Urstöger[1,2], Andreas Steinegger[3], Robert Schennach[2,4], Ulrich Hirn[1,2]***

**1** Institute for Paper, Pulp and Fiber Technology, Graz University of Technology, Graz, Austria, **2** CD-Laboratory for Fiber Swelling and Paper Performance, Graz University of Technology, Graz, Austria, **3** Institute of Analytical Chemistry and Food Chemistry, Graz University of Technology, Graz, Austria, **4** Institute of Solid State Physics, Graz University of Technology, Graz, Austria

* ulrich.hirn@tugraz.at

**Data Availability Statement:** All relevant data are within the paper and its Supporting Information files.

## Abstract

Two molecules, 7-(diethylamino)coumarin-3-carbohydrazide (DCCH) and fluorescein-5-thiosemicarbazide (FTSC) were investigated in different solvents, under varying pH conditions regarding their spectroscopic properties for the usage as a Förster Resonance Energy Transfer (FRET) pair to study the molecular interaction between cellulosic surfaces. All the relevant spectroscopic properties to determine the Förster distance were measured and the performance as a FRET system was checked. From the results, it is clear that the environmental conditions need to be accurately controlled as both, but especially the FTSC dyes are sensitive to changes. For high enough concentrations positive FRET systems were observed in DMF, DMSO, $H_2O$, THF and alkaline DMF. However due to the low quantum yield of the unmodified DCCH throughout the investigated parameter range and the strong environmental dependency of FTSC, both dyes are not preferable for being used in a FRET system for studying interaction between cellulosic surfaces.

## Introduction

Förster Resonance Energy Transfer (FRET) is a technique mainly used to determine the distance between a so-called donor and an acceptor molecule. Developed by Theodor Förster in the 1940s, FRET has evolved to a standard investigation technique in cellular biology to study the interactions between protein molecules, in the usage of biomarkers or for building sensors. [1–3]

The theory of the physical effect is based on the electromagnetic interaction between a donor and an acceptor molecule. An incident photon excites the donor molecule which can transfer its energy, by a non-radiative interaction, to the acceptor. In principle, there are other interaction mechanisms but Förster could show that in the near field range (approx. 1–20 nm) the dominant energy transfer mechanism is Förster transfer. The efficiency of the transfer depends on the proximity of the two molecules and is in principle determined by the Förster Radius ($R_0$). This quantity is specific for every Donor-Acceptor pair and gives the range within

**Funding:** The authors thank the Austrian Federal Ministry of Economy, Family and Youth and the Austrian National Foundation for Research, Technology and Development for their financial support. Also the industrial partners Mondi, Canon Production Printing, SIG Combibloc and Kelheim Fibres are acknowledged.

which the distance can be quantified. Beyond this range, which is about ½ $R_0$–2 $R_0$, it may still be possible to qualitatively determine whether the molecules do or do not interact. Closer than the lower limit the energy transfer mechanism is called Dexter transfer.[4,5]

While many people already use this technique as a standard analyzing tool in life sciences, others strive to develop further applications.[6,7] However, in all cases it is crucial to have a deep and thorough understanding of the employed dyes and the system that is under investigation. We present here a detailed analysis of 7-(diethylamino)coumarin-3-carbohydrazide (DCCH) and fluorescein-5-thiosemicarbazide (FTSC) in different solvents, under varying pH conditions regarding their spectroscopic properties for the usage as a FRET pair. The two chromophores were used by Thomson et al. to detect an interaction between paper fibers. [8,9], which is a promising approach to quantify the nanometer-scale contact area available for adhesion between the fiber surfaces [22].

The following brief introduction to the most prominent equations in Förster Theory were taken from chapter 3 of the book, FRET–Förster Resonance Energy Transfer, From Theory to Applications, written by Prof. Van der Meer and is based partly on the original papers of Förster.[3,10,11] The efficiency of the energy transfer ($\eta_{eff}$ [–]) is given by Eq (1):

$$n_{eff} = \frac{1}{1 + \left(\frac{r}{R_0}\right)^6} \tag{1}$$

Where r [nm] stands for the distance between the donor and acceptor and $R_0$ [nm] stands for the Förster radius of the donor-acceptor pair. The Förster radius or distance can be calculated via Eq (2):

$$R_0^6 = \frac{9\ln 10}{128\,\Pi^5 N_A}\left(k^2 n^{-4} Q_0 J\right) \tag{2}$$

Where $N_A$ [mol$^{-1}$] is Avogadro's constant, k [–] is the orientation factor, n [–] is the index of refraction of the medium, $Q_0$ [–] is the quantum efficiency of the donor in the absence of FRET and J [M$^{-1}$cm$^{-1}\lambda^4$] is the overlap integral which is calculated with Eq (3):

$$J = \int f_D(\lambda)\epsilon_A(\lambda)\lambda^4 d\lambda \tag{3}$$

Where $f_D$ [a.u.] is the area normalized fluorescence intensity of the donor, $\epsilon_A$ [M$^{-1}$cm$^{-1}$] is the attenuation coefficient of the acceptor and $\lambda$ [nm] is the wavelength.

By measuring the transfer efficiency one can calculate the distance between the Donor and Acceptor once the Förster radius for the system has been determined. Practically, this can be done by different methods which can be implemented in either microscopy setups or measured by spectrophotometry, which was used in this paper. Regardless of the measurement method, a FRET signal can be detected by different aspects of the effect. Two prominent ones are the donor quenching (data provided in the S1 File) which measures the decrease of the donor fluorescence due to FRET; or the acceptor sensation method which measures the increase of the acceptor fluorescence due to FRET. While donor quenching is an indication for FRET, one cannot be certain of it as there are other mechanisms such as concentration quenching that can deactivate the excited Donor. Acceptor sensation on the other hand provides a compelling argument for FRET as the acceptor fluorescence can only be increased by some sort of energy transfer. To be certain of the resulting data many people have developed equations that correct the efficiency for all possible cross talk situations.[12,13] To quantify the FRET efficiency using a spectrophotometer it is necessary to spectrally unmix the detected

emission curves. Then the FRET efficiency η [–] can be calculated by

$$\eta_{\text{eff}} = \left( \frac{I_{AD}}{I_A} - 1 \right) * \frac{\epsilon_{Acceptor}}{\epsilon_{Donor}} \tag{4}$$

Where $I_{AD}$ [a.u.] and $I_A$ [a.u.] are the fluorescence emission intensities of the acceptor in the presence and in the absence of donor, respectively. $\epsilon_{Acceptor}$ [$M^{-1}cm^{-1}$]and $\epsilon_{Donor}$ [$M^{-1}cm^{-1}$] are the molar attenuation coefficients of the acceptor and the donor molecule at the used excitation wavelength, respectively.

The fluorescence quantum yield of a molecule is the probability of an excited state to be deactivated by fluorescence rather than by another, non-radiative mechanism. The quantum yield can be measured via the usage of a known fluorescence standard by Eq 4,

$$Q_{F(x)} = Q_{F(s)} \frac{A_s}{A_x} \frac{F_x}{F_s} \frac{n_x^2}{n_s^2} \tag{5}$$

where $Q_F$ [–] stands for the fluorescence quantum yield, A [OD] is the absorbance at the excitation wavelength, F [a.u.] is the area under the corrected emission curve and n [–] is the refractive index of the medium. The indices x and s refer to the unknown and the standard sample, respectively. Another possibility to measure the quantum yield is with the absolute method. [14–18]

The molar attenuation coefficient ($\epsilon$ [$M^{-1}cm^{-1}$]) is connected to the absorbance by Beer Lamberts Law (Eq 6)

$$A = \epsilon\, c\, l \tag{6}$$

Where A [OD] is the absorbance defined as the negative decadic logarithm of the measured transmittance, c [mol/L] is the concentration of the solution and l [cm] is the length of the light path. Correctly measured the attenuation coefficient tells you how well a substance absorbs light at a certain wavelength independent of concentration or geometrical considerations.

This paper focuses on the stepwise determination of spectral properties (excitation/emission spectra, attenuation Coefficient, quantum yield) needed to calculate the Förster radius and the subsequent investigations of mixtures to determine whether an energy transfer can be observed or not, and under which conditions the system works well. Ultimately this method will be adapted for measuring adhesion between surfaces and to understand a complicated system like that groundwork such as this is necessary. For a related system it was shown that a good FRET response was achievable.[10,11]

## Experimental

### Materials

The dyes 7-(diethylamino)coumarin-3-carbohydrazide (DCCH, Purity 95%, CAS: 100343-98-4), fluorescein-5-thiosemicarbazide (FTSC, Purity 99%, CAS: 76863-28-0) and the standard fluorescein sodium salt were bought from Santacruz Biotechnology (Dallas, Texas, USA). Coumarin 30 was bought from (Sigma Aldrich) The solvents N,N-dimethylformamide (DMF), dimethyl sulfoxide (DMSO), Tetrahydrofuran (THF), and Acetonitril were purchased from VWR (Vienna, Austria). All chemicals were used without further purification.

## Sample preparation and methods

The volume of the cuvette for all measured solutions was 2 ml. In the case of the capillary a volume of 2 ml was prepared and thoroughly mixed. To investigate the influence of the pH value on the properties the pH of the solutions was adjusted by adding 12 μL triethylamine to the 2 ml solution of $H_2O$, DMF and DMSO. To accordingly adjust the pH in THF 10-times the volume of triethyamine was used (120 μL). The pH values of the systems are reported in the S1 File.

The absorbance was measured with a VARIAN CARY, UV-vis spectrophotometer. To minimize the inner filter effect and deviations from Beer-Lamberts law the optical density of the transmission measurements never exceeded 0.5 OD. Fluorescence spectra were recorded on a FluoroLog 3 spectrofluorometer from Horiba Scientific equipped with a R2658 photomultiplier from Hamamatsu.

FRET experiments were performed with concentrations of molecules of 0.1 mM and 1 mM in two different ways. For the lower concentration experiments were conducted using standard optical glass 10 mm precision cuvettes from Hellma Analytics, with the Fluorolog. Additionally, FRET experiments were conducted using a capillary due to the need to go to higher concentrations. The spectra of those experiments were measured with a RF-5301PC, spectrofluorophotometer from Shimadzu. Using a capillary was necessary to avoid the inner filter effect which occurs at higher concentrations. All FRET experiments were performed using a ratio of 1:1 between the molecules. Equipment used for Fluorometry or UV/Vis was spectroscopy grade and cleaned rigorously between measurements. Also, care was taken to minimize the exposure of the solutions to ambient light to avoid photobleaching.

As a reference for the quantum yield measurements coumarin 30 in acetonitril and fluorescein sodium salt in 0.1 M NaOH were used with reported quantum yields of 0.55 and 0.95, respectively.[19,20] The quantum yields for the fluorescein compound in DMF, DMSO and THF were determined by the absolute method using a Quanta-ϕ integrating sphere-based setup from Horiba Scientific connect to the Fluorolog 3 spectrophotometer. The concentration of the solutions was chosen such that the absorbance in the UV-vis is as large as close as possible but still below 0.05 OD ($\lambda_{max} \leq 0.05$ OD). This way one can be sure that the inner filter effect does not influence the result. The method was taken from Würth et al. [18]

Since the dyes should eventually be used embedded in a matrix or bound to cellulosic surfaces measurements of the quantum yields of the dyes in Poly(2-hydroxyethyl methacrylate) (pHema thin films) and chemically bound to paper fibers (paper sheets) were also included. The production of the thin films and the dyeing of the paper fibers can be found in Urstöger et al.[21] The quantum yields were measured by the absolute method either directly of the thin films or of a small sheet of paper produced with alkaline (pH 9, NaOH) water.

To test if a modification of the dyes can improve the QY, DCCH was modified using an acetylation reaction. DCCH was dissolved by weighting 1 mg of DCCH into 500 μl of THF. Subsequently 10 μl of Triethylamine and 10 μl acetly chloride were added to the solution. Thin layer chromatography (TLC) was used to determine if the reaction was finished. The solution was filtered with a Chromafil XTRA PTFE 0.45 μm filter. Before measuring the QY of the solution the pH was adjusted using TEA to match the unmodified version.

## Results and discussion

### Molar attenuation coefficient

As a key spectroscopical property, the attenuation coefficient tells you how well a substance absorbs light at a certain wavelength. On the one hand this quantity is important as it is

directly connected to the Förster Radius (Eq 2) by the overlap integral (Eq 3) which in the end determines at which distances the method will be applicable. On the other hand in combination with the quantum yield, the molar attenuation coefficient also determines how well the donor will provide energy for the transfer process. This means that for the acceptor a high attenuation coefficient is beneficial as the excitation of this energetic transition becomes more likely. Generally a higher molar attenuation coefficient is desirable however, under some circumstances it could be beneficial to have a lower coefficient e.g. in the case where a high concentration of molecules is needed but due to the high molar attenuation coefficient the solution becomes hard to measure due to the inner filter effect. In such a case a lower molar attenuation coefficient could be better. Therefore, this property is not only important for the calculation of the Förster radius but is also very relevant as a property itself.

As can be seen in Fig 1 the coefficient can vary quite drastically depending on the solvent as well as on the pH value. In the case of DCCH, one can see in Fig 1A that, without any base, the peak of the absorption band shifts with different solvents ranging from 412 to 427 nm. They also shift in intensity up to a factor of 2.5. When a base is added (Fig 1B) the shift in position becomes smaller and the spectra become more similar. The intensity in DMF and THF gets smaller when adding base while the intensity in DMSO stays almost constant and in $H_2O$ it increases slightly. The maximum of the spectra shifts towards higher wavelengths by 5–13 nm. The FTSC curves in Fig 1C and 1D can be analyzed in a similar fashion. Without any base the attenuation coefficient of FTSC in all solvents except for water is extremely low. In water, we still get a sufficient absorbance while in other solvents it is almost zero throughout the complete spectrum. However, when a base is added to the system the intensity increases strongly while the position of the maximum and the shape of the curve also changes. The shift in position is quite high and lies between 47–56 nm. This trend is also seen by others and is likely to be attributed to variations in the protonation states of fluorescein.[22,23] In $H_2O$ the maximum coincides with the NaOH reference while the DMF and DMSO samples show a distinct redshift. Adding the same amount of base to THF did not yield any change and even the 10-fold amount resulted only in little change. An important feature of these graphs is the increasing attenuation coefficient of FTSC in both, neutral and alkaline, conditions from 300 to 350 nm. The measurement in this region is tricky as the solvents start to absorb in that range as well. However, we believe that this is a real feature because an excitation in this range also appears in the fluorescence measurements.[23,24]

## Quantum yield

The quantum yields were determined by the relative method and the absolute method described in the methods. A high quantum yield of the donor is worth striving for as it gives the donor the opportunity to provide more energy for the transfer process. Also, it is directly connected to the Förster distance via Eq 2. As can be seen in Table 1 the quantum yield can vary significantly depending on solvent and pH value. However, the QY of DCCH was generally low compared to FTSC. To check if the reason for the low QY of the molecules is the hydrazide group a chemical modification of DCCH was performed. The molecule was modified by adding acetyl chloride and triethylamine to the THF solution which results in an acetylation of the hydrazide group to create a hydrazine. The reaction scheme can be seen in Fig 2. As can be seen in Table 1 the quantum yield increased by more than a factor of 4. The QY for the fluorescein compound in DMF, DMSO and THF were measured absolutely and can be seen in Table 1. In all three solvents, the QY becomes very low most likely due to the reason that fluorescein is present in its lactone form which is known for having a very low QY. [25] Additionally in Table 1 the QY of the dyes incorporated in different structures like Poly

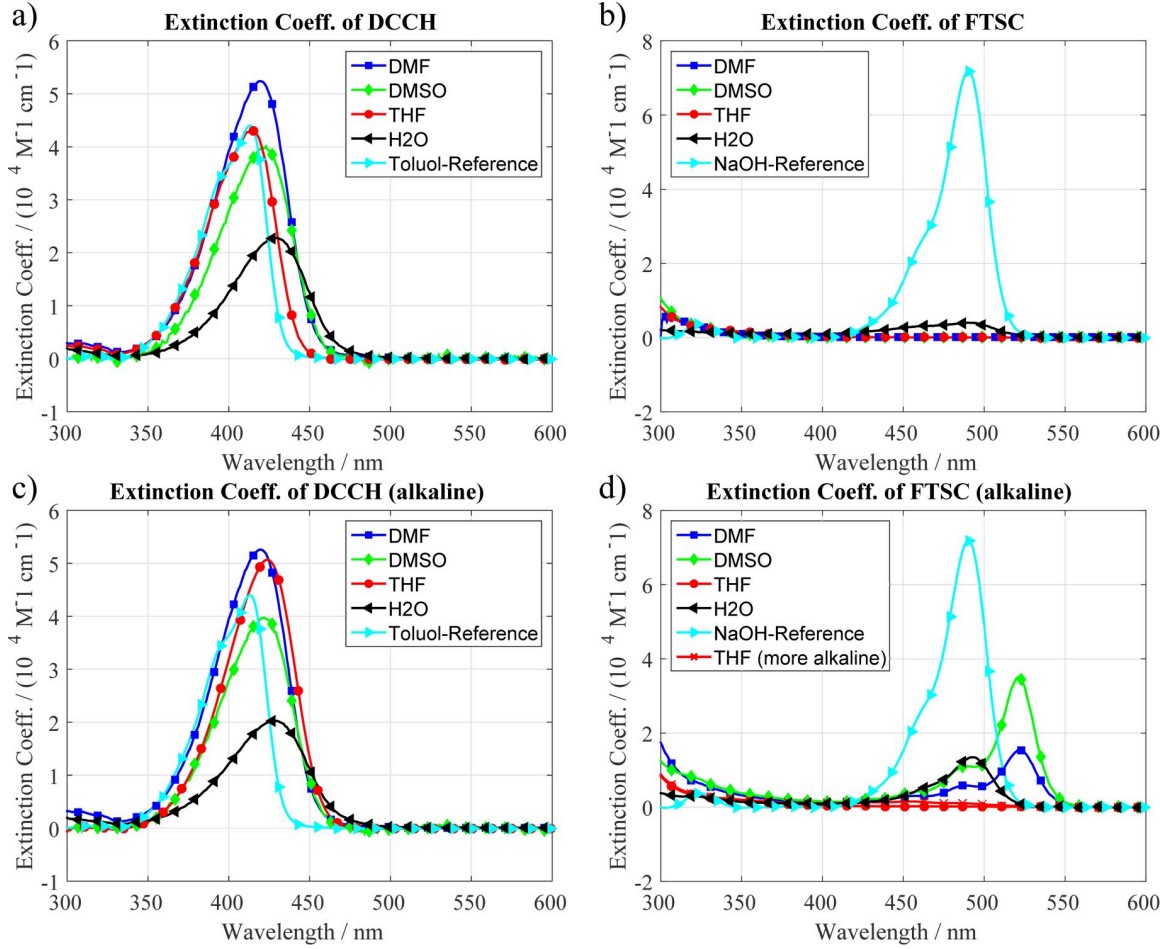

**Fig 1. Molar attenuation coefficient of DCCH and FTSC at different pH conditions and in various solvents. a,b)** Attenuation Coefficient of DCCH. The coefficients vary up to a factor of approx. 2.5 in the various solvents but do not show much change due to pH except for THF. The Toluol labeled sample represents the measurement of the reference molecule Coumarin 30 for the quantum yield (QY) measurements. **c,d)** Attenuation coefficient of FTSC. NaOH is the measurement of Fluorescein Sodium Salt needed as a reference for the QY of FTSC. **c)** One can see that the absorbance of FTSC is almost fully quenched in DMF, DMSO and THF in the visible range. In $H_2O$ the performance is a little better. **d)** Going to alkaline conditions increases the attenuation coefficient in most solvents.

(2-hydroxyethyl methacrylate) (pHema) and paper fibers was investigated. In the case of pHema the dyes were physically mixed in alkaline conditions into the pHema matrix. In the case of DCCH this increases the QY significantly by almost a factor of 5 whereas it decreases for the FTSC. An increase of quantum yields of dyes has been reported due to the immobilization of chemical groups.[26] The FTSC appears to be slightly quenched in the matrix. In the

**Table 1. Quantum yield [-] of DCCH and FTSC for different solvents and pH value.** The QY was determined by a relative measurement. Repeated measurements yielded an error of 20%.

| Solvent / Dye | $H_2O$ | DMF | DMSO | THF | THF modified | pHema | Paper Fibers |
|---|---|---|---|---|---|---|---|
| DCCH | 0.01 | 0.01 | 0.00 | 0.04 | - | - | - |
| DCCH alkaline | 0.01 | 0.02 | 0.01 | 0.04 | 0.19 | 0.14 | 0.23 |
| FTSC | 0.51 | 0.23 | 0.16 | 0.20 | - | - | - |
| FTSC alkaline | 0.23 | 0.26 | 0.10 | 0.29 | - | 0.12 | 0.41 |

**Fig 2. Chemical modification of the hydrazide group by acetylation with acetyl chloride.** The resulting hydrazine increases the QY of the Coumarin by a factor of 5.

case of the paper fibers the dyes were chemically bonded into the fiber matrix which lead to a strong increase in the QY which is likely a combination of the increase due to modification of the molecule and the immobilization effect described earlier. The dyeing is explained in the paper of Urstöger et al. [21]

## Fluorescence spectra

Further important spectroscopic quantities for FRET are the fluorescence excitation and emission spectra of the molecules. They can be seen in Figs 3 and 4. Under the condition that the excitation spectra of the respective acceptor do not vary too much from the attenuation coefficient one can here nicely see the spectral overlap that is a necessary condition for FRET as seen in Eqs 2 and 3.

In Fig 3A and 3C the spectra of DCCH and FTSC in $H_2O$, in neutral, and in alkaline conditions are plotted. Compared to the attenuation Coefficient the excitation spectra of DCCH are in both cases shifted to the blue by about 10 nm. For FTSC, the maxima of the excitations coincide with the maxima of the attenuation Coefficients. In Fig 3C and 3D the spectra of the molecules in THF can be seen. The most prominent feature in this data is the strong blue shift of the FTSC excitation and emission spectrum. In THF the EX/EM peaks appear at 330 and 390 nm, respectively. This leads to a switch in the roles concerning a FRET application. In neutral THF the FTSC, shifts so strongly to the blue that it now takes on the role of the donor while the DCCH stays roughly the same and becomes the acceptor. The resulting spectra of FTSC in alkaline THF were low in intensity which is the reason why they appear quite noisy in the normalized graphs. We believe this is mainly due to the very low attenuation coefficient. Also the systems exhibit many excitation peaks which are likely due to varying amount of fluorescein in different protonation states.[23] The excitation spectra of DCCH in THF are consistent with the attenuation coefficient measurements of Fig 1. In Fig 4 the spectra of the molecules in DMF and DMSO can be seen. Due to the similarity of the solvents regarding many of their

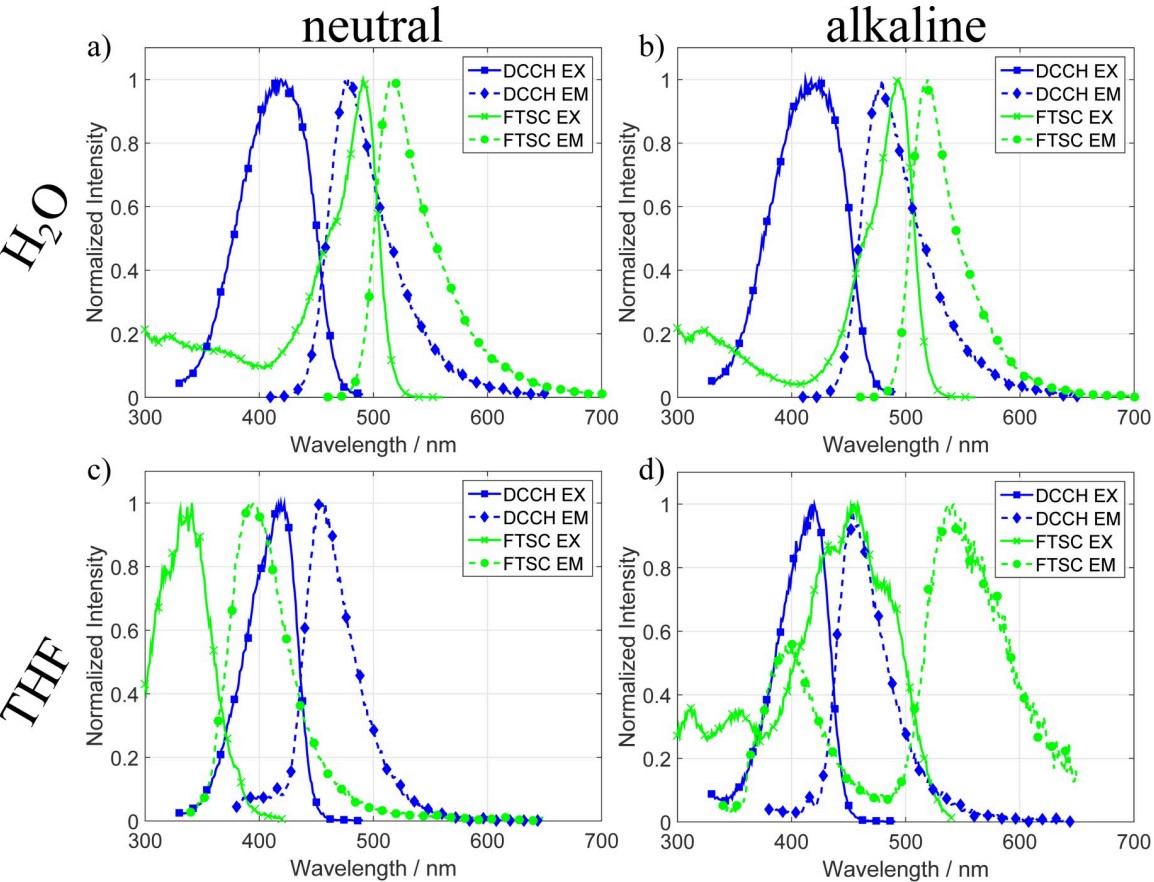

**Fig 3. Normalized excitation and emission spectra of DCCH and FTSC in different solvents and pH values.** In **a)** and **b)** one can see the spectra in water under neutral and alkaline conditions. One important feature in **c)** and **d)** is that in the neutral solution the FTSC spectra shift strongly (130–150 nm) into the blue compared to the H$_2$O spectra. In the alkaline solution the spectra shift back to the red and exhibit many side excitations.[22].

properties, the investigated spectra look quite similar. However, compared to H$_2$O and THF there are many differences. First one can see in Fig 4A and 4C the same trend as in THF, namely that the FTSC EX/EM shifts so far to the blue that it takes on the role of acceptor. Compared to each other the excitation and emission spectra of FTSC in DMF and DMSO in neutral as well as in alkaline conditions look very similar. With the one difference that in DMF, FTSC exhibits an additional emission at 410 nm. The excitation spectra also look very similar to each other and also correlate to the attenuation coefficient measurements. DCCH shows a different behavior. First, the emission spectrum of the molecule is only weakly affected by the solvent or the pH conditions of the system. Second, the excitation changes quite strongly. Especially in DMF the excitation spectrum exhibits suddenly two excitations at 370 and 450 nm from which the 450 nm excitation is also much narrower than all the other recorded ones. In alkaline conditions (Fig 4B and 4D) both spectra exhibit an excitation peak at approx. 390 nm while the emission maximum is around 460 nm.

One explanation for the shifting spectra of FTSC is again the well-known pH dependence of the molecule.[23] Different protonation levels shift the electronic structure in such a way that a different transition becomes more probable. This is especially visible in the cases of THF, DMF and DMSO in which the shift of the maxima ranges up to 180 nm. Another effect on the spectra is caused by the interaction with the solvents themselves. Depending on the

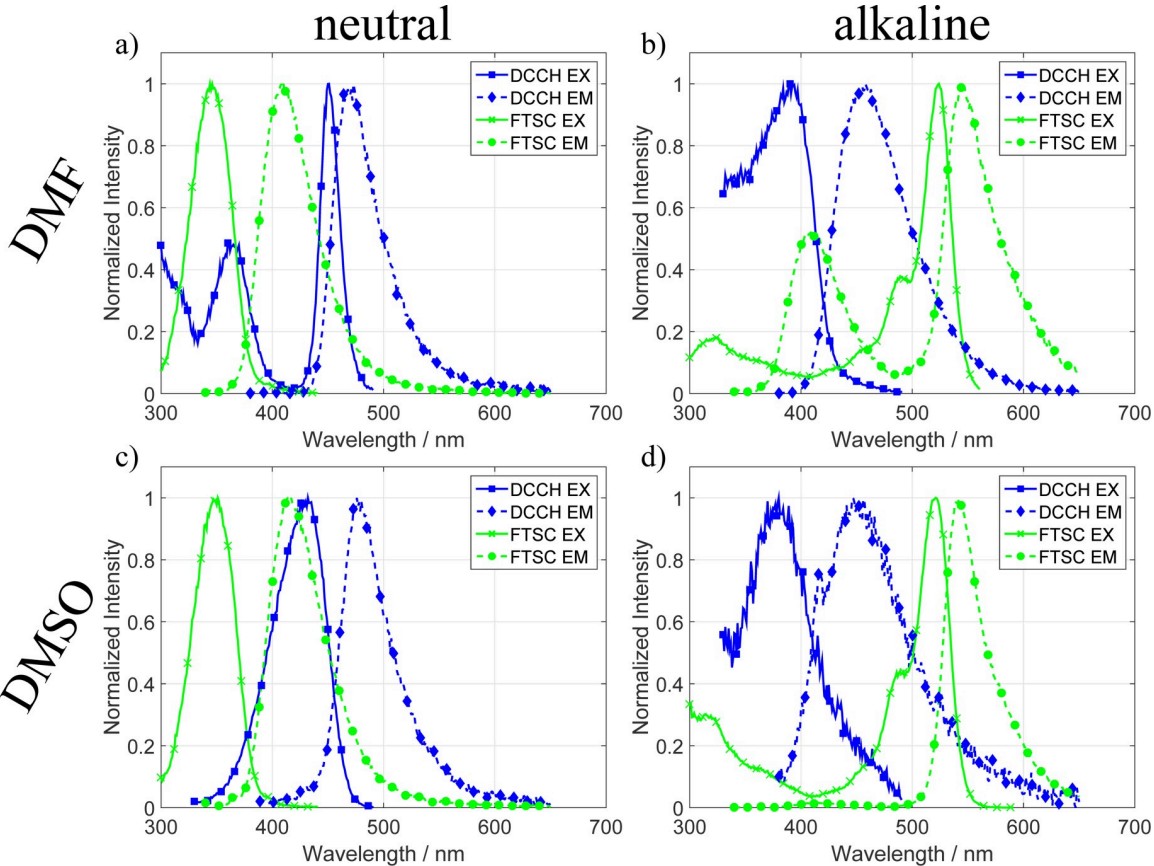

**Fig 4. Normalized excitation and emission spectra of DCCH and FTSC in different solvents and pH values.** In **a)** and **b)** one can see the spectra in DMF. Here the FTSC EX/EM shifts to 340/410 nm in the neutral case and shifts back to higher wavelengths (520/540 nm) in alkaline conditions where it additionally exhibits a second emission at 410 nm. The DCCH main excitation becomes narrower and a second excitation at 370 nm appears in the neutral state. Going to alkaline conditions shifts the DCCH spectra into the blue. **c)** and **d)** show the spectra in DMSO. The spectra for FTSC look quite similar as in DMF with the exception of the missing additional emission at 410 nm in the alkaline state. The DCCH spectra for the alkaline conditions are similar while in the neutral state the excitation spectrum shifts a few nm into the blue and the spectra become wider.

polarity and the strength of the interaction of the solvent molecules with the dyes the peaks can shift, change their width or even exhibit additional peaks. The combination of the two investigated dyes in one solvent determines to a large extent whether the dyes are compatible when they are used as a FRET pair.

## Förster distance

Having all the necessary data for the calculation measured, the Förster distances given by Eq 2 depending on pH and solvent were calculated. As can be seen in Table 2 the radius can range from 1.82 to 3.10 nm. The different radii can be put into two groups depending on which molecule, DCCH or FTSC, takes on the part of the donor in the FRET pair. In Table 2, this has

**Table 2. Förster distance in nm of the DCCH and FTSC pair in dependence of solvent and pH value.** The D and the F in the Table represent whether DCCH or FTSC took the role of the Donor in the combination, respectively. The other molecule was the acceptor.

| Solvent<br>pH | $H_2O$ | DMF | DMSO | THF |
|---|---|---|---|---|
| Neutral | D 2.5 ± 0.6 | F 3.1 ± 0.7 | F 2.9 ± 0.7 | F 2.9 ± 0.7 |
| With base | D 2.9 ± 0.7 | D 3.0 ± 0.7 | D 2.6 ± 0.6 | D 1.8 ± 0.5 |

been marked with a D or an F depending whether DCCH or FTSC takes the part of the Donor, respectively. The variation in the Förster radius has an influence on the application of a chosen FRET pair to investigate the interaction between molecules. Depending on the solvent and on the pH value the distance dependence of the Förster transfer will be different. This means that a chosen FRET pair might work in one solvent or pH but fail under changing conditions. This is true whether FRET is qualitatively or quantitatively used to investigate a system. Qualitatively, this means that an interaction might not be detected due to a too far distance. If FRET is used to quantitatively study the interaction between molecules a changing Förster distance needs to be considered when performing the calculations as it can be a source of error.

### FRET system

In this section we show under which of the previously discussed systems we were able to observe a Förster transfer and which conditions did not. The key difference in the two experimental series seen in Figs 5 and 6 was the concentration of the dyes in the solvents. In the first trial (Fig 5) all the measurements were performed with concentrations of 0.1 mM. With these concentrations we were only able to get a FRET signal using DMF and DMSO.

As mentioned in the experimental section it is possible to go to higher concentrations while avoiding the inner filter effect by using a capillary. By doing this we were able to detect in 3 more of the 8 investigated systems a FRET signal. The positive systems can be seen in Fig 6.

In total, we were able to get a FRET signal in 5 of the 8 investigated systems. The quantified parameters can be seen in Table 3. Interestingly 3 of the 5 positive systems, namely in THF, DMF and DMSO, were achieved with the FTSC molecule taking on the role of the Donor although the initial thought was to use it as the acceptor. It was also under these conditions that the FRET efficiency was the highest. A Table with the performed calculations of the FRET efficiencies can be found in the S1 File. By comparing the values in Table 3 it becomes obvious that the highest FRET efficiencies were achieved when both, the quantum yield of the donor and the molar attenuation coefficient of the acceptor, were large.

One of the most compelling techniques to investigate FRET systems would be fluorescence lifetime imaging (FLIM) which measures the FRET efficiency by the decrease of the characteristic lifetime the donor has due to the presence of acceptor molecules. Unfortunately, the

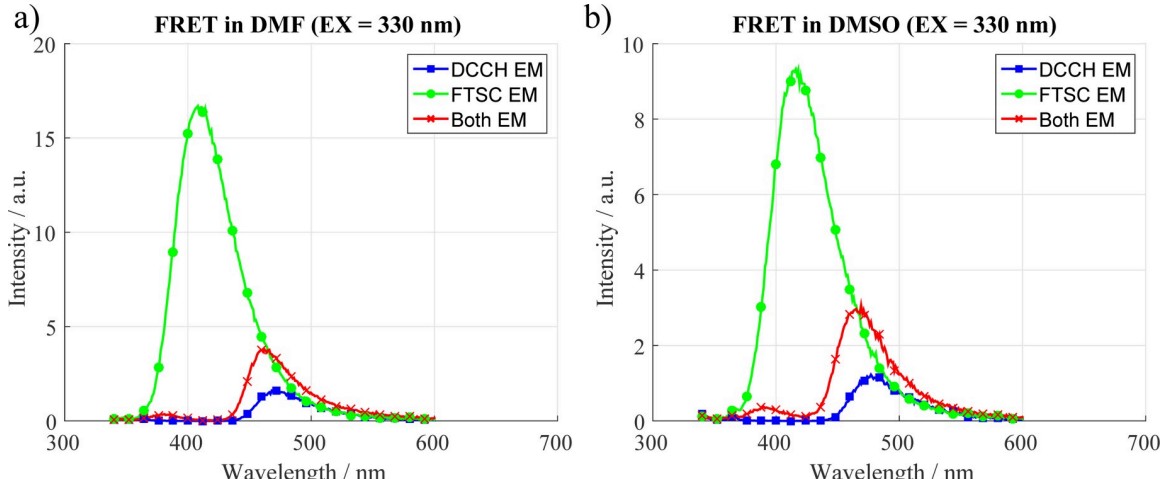

**Fig 5. Fluorescence measurements with a concentration of 0.1 mM.** When the dyes are mixed together the FTSC fluorescence is quenched and the DCCH fluorescence is enhanced which is a clear sign of FRET.

authors did not have access to FLIM although we encourage future readers to use the technique additionally to others as it is such a valuable tool to investigate the FRET response of a system. Due to the underlying investigation it was possible to develop an already published method to investigate interactions between surfaces by measuring the adhesion between pHema thin films dyed with DCCH and FTSC. [21]

## Conclusions

In this work, 7-(diethylamino)coumarin-3-carbohydrazide (DCCH) and fluorescein-5-thiosemicarbazide (FTSC) were investigated regarding their respective spectral properties in different solvents and under changing pH conditions with the goal of finding an optimal Förster Resonance Energy Transfer (FRET) pair. DCCH and FTSC do not appear to perform well in FRET experiments. On the one hand this is due to the strong dependence of the environment of the FTSC and on the other hand due to the low quantum yield of DCCH. Additionally, the FTSC is almost completely quenched (molar attenuation coefficient) when dissolved in DMF, DMSO or THF. After a chemical modification of DCCH it could be shown that the quantum yield improves significantly which leads to a better dye performance. The Förster Radii for different environmental conditions were determined and it was shown that they are slightly affected by solvent and pH. Therefore, for measuring and quantifying FRET one needs to carefully and meticulously determine and control the environmental influences and conditions under which

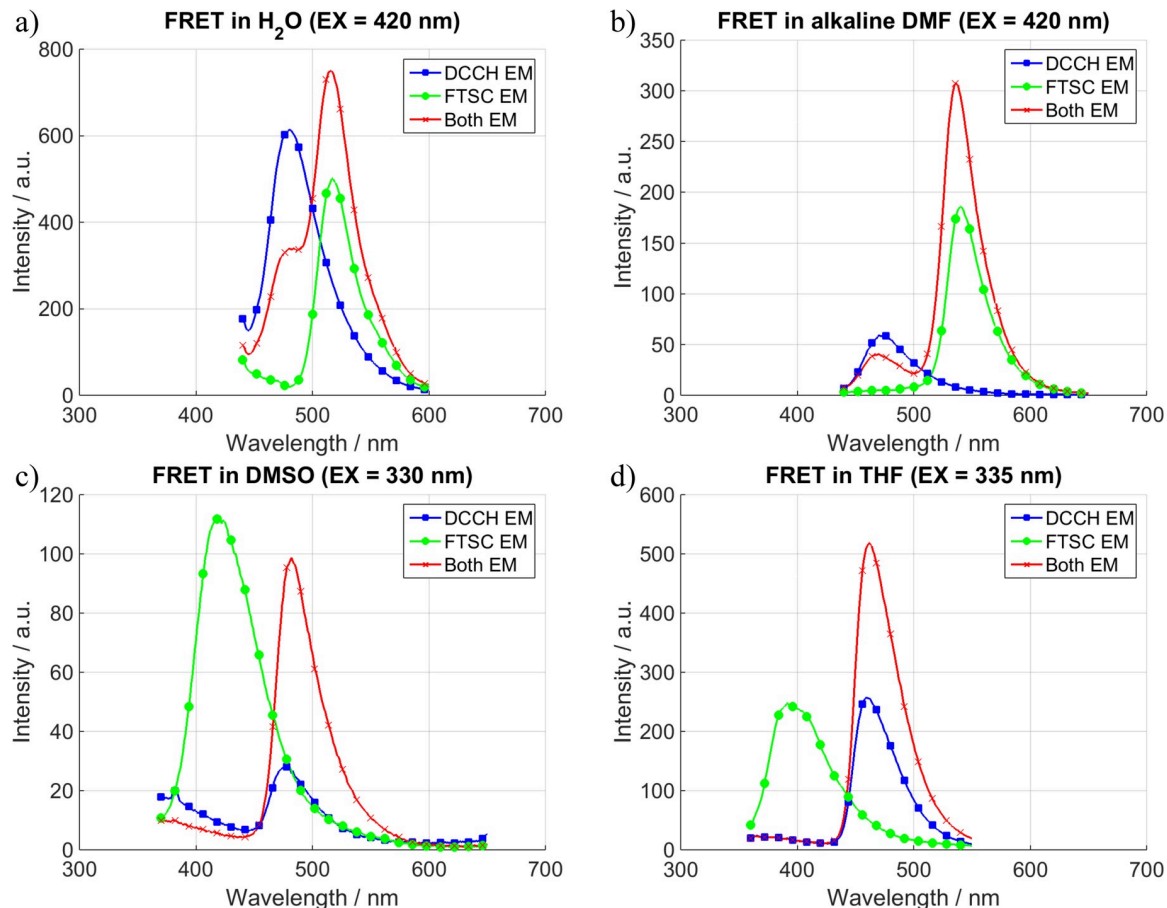

**Fig 6. FRET measurements using a capillary and thus going to higher concentrations.** Due to the higher concentration a FRET signal is also detected under other conditions. (Conc. $H_2O$ = 0.15 mM, conc. Other = 1 mM).

**Table 3. Comparison between relevant parameters and the resulting Fret signal.** The reported attenuation coefficient is the area under the measured curve and the provided FRET efficiency stems from acceptor sensitation (Eq 4). The results for donor quenching can be found in the S1 File.

| Solvent | Donor Molecule | Attenuation Coefficient Acceptor [M^-1 cm^-1] | QY Donor [-] | $R_0$ [nm] | FRET Efficiency / [%] |
|---|---|---|---|---|---|
| $H_2O$ | DCCH | 3.1E+5 | 0.01 | 2.5 | 3 |
| $H_2O$ alkaline | DCCH | 5.9E+5 | 0.01 | 2.9 | 0 |
| DMF | FTSC | 2.9E+6 | 0.23 | 3.1 | 73 |
| DMF alkaline | DCCH | 7.2E+5 | 0.02 | 3.0 | 2 |
| DMSO | FTSC | 2.2E+6 | 0.16 | 2.9 | 15/22* |
| DMSO alkaline | DCCH | 1.4E+6 | 0.01 | 2.6 | 0 |
| THF | FTSC | 2.3E+6 | 0.2 | 2.9 | 9 |
| THF alkaline | DCCH | 2.5E+4 | 0.04 | 1.8 | 0 |

*15/22 corresponds to the efficiency in the lower (0.1 mM) and higher (1 mM) concentrated solutions, respectively.

one wants to measure an interaction. Finally, it was observed in three cases that the initially intended donor (DCCH) and acceptor (FTSC) switched their role in the case of certain environmental conditions (DMF, DMSO, THF). In these cases, the FRET efficiency was the highest because these systems had the highest quantum yield of the donor and the highest molar attenuation coefficient of the acceptor. These investigations of the FRET systems in solution were later adapted to measure adhesion between pHEMA films. [21]. Cellulosic surfaces, however, are developing full adhesion only when they are in contact during drying from the water swollen state[27]. The investigated system therefore is, due to the instability of its fluorescence characteristics in water, only partly suited to study the adhesion between cellulosic surfaces.

## Supporting information

**S1 File.**
(DOCX)

## Author Contributions

**Conceptualization:** Ulrich Hirn.

**Funding acquisition:** Ulrich Hirn.

**Investigation:** Georg Urstöger, Andreas Steinegger, Robert Schennach, Ulrich Hirn.

**Methodology:** Andreas Steinegger, Robert Schennach, Ulrich Hirn.

**Supervision:** Robert Schennach, Ulrich Hirn.

**Writing – original draft:** Georg Urstöger, Robert Schennach, Ulrich Hirn.

**Writing – review & editing:** Georg Urstöger, Robert Schennach, Ulrich Hirn.

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
