## [Decision Letter · Decision Letter 0]

27 Nov 2019

PONE-D-19-23740

Spectroscopic Investigation of DCCH and FTSC as a potential pair for Förster Resonance Energy Transfer in different solvents

PLOS ONE

Dear Prof. Hirn,

Thank you for submitting your manuscript to PLOS ONE. After careful consideration, we feel that it has merit but does not fully meet PLOS ONE’s publication criteria as it currently stands. Therefore, we invite you to submit a revised version of the manuscript that addresses the points raised during the review process.

Given that reviewer 2 recommended rejection and had a number of specific comments, I would invite you to carefully consider these in your response.  

We would appreciate receiving your revised manuscript by Jan 11 2020 11:59PM. To enhance the reproducibility of your results, we recommend that if applicable you deposit your laboratory protocols in protocols.io, where a protocol can be assigned its own identifier (DOI) such that it can be cited independently in the future. For instructions see: http://journals.plos.org/plosone/s/submission-guidelines#loc-laboratory-protocols

We look forward to receiving your revised manuscript.

Kind regards,

Warren Batchelor

Academic Editor

PLOS ONE

Reviewers' comments:

Reviewer's Responses to Questions

**Comments to the Author**

1. Is the manuscript technically sound, and do the data support the conclusions?

Reviewer #1: Yes

Reviewer #2: No

2. Has the statistical analysis been performed appropriately and rigorously? 

Reviewer #1: N/A

Reviewer #2: No

3. Have the authors made all data underlying the findings in their manuscript fully available?

Reviewer #1: Yes

Reviewer #2: No

4. Is the manuscript presented in an intelligible fashion and written in standard English?

Reviewer #1: No

Reviewer #2: No

5. Review Comments to the Author

Reviewer #1: The MS entitled, "Spectroscopic Investigation of DCCH and FTSC as a potential pair for Förster Resonance Energy Transfer in different solvents" by Ulrich Hirn et al., presents experimental evaluation of two molecules FTSC and DCCH in various solvent as function of varying pH and the results based on steady state absorption and fluorescence spectroscopic measurements.

The Introduction section is quite long and needs to be trimmed. The paragraph (page 3; lines #81-86) may appropriately be shifted to the end of Introduction section.

In eqn 5, the attenuation coefficient epsilon is not described in the following text, as it appears for the first time.

In Results, on page 8; lines # 232-235, the sentence needs to be revised as it is unable to provide correct meaning.

Similarly, on page 9; lines 281-283, the sentence be revised to give the reader correct meaning.

Table 2 on page 10, should mention whether the values provided are absolute or % values.

On the same page, lines # 313-314 the sentence "Another important ...... spectra of molecules" is incorrect both grammatically as well as scientifically. Rewrite it.

On page 10, paragraph (lines #407-412), should consider the contribution of radiative transfer as the distance of approx. 16.4 nm (quite greater than 2xForster distance) is also able to show FRET signal.

In Table 4, the authors may give the values of epsilon, QY, R0 instead of giving symbols to show their strengths.

Please clarify whether, the intensity values mentioned i Fig. 7 are in rel unite or counts.

While the results are interesting, the discussion of the same needs to be presented with more lucidity. The quality of the paper would greatly enhance if the authors include the time resolved measurements (fluorescence lifetimes) also which confirm the resonance energy transfer process in such mixtures.

The authors are advised to take the assistance of native English speaker to improve the language of the MS. As it is, there are several grammatical errors, punctuation errors and incorrect sentences leading to unclear information. H2O is mentioned at several places without proper use of subscript. Whole MS may be checked for such mistakes. The paper may be accepted after appropriately revising it.

Reviewer #2: The manuscript describes the spectroscopic analysis of two dyes in a variety of solvents that have been used as a FRET pair in a previous publication to measure the contact between paper fibers. The manuscript focuses mostly on the spectroscopic properties of the molecules in solvents. Significant issues with the manuscript exist that reduce its clarity, reproducibility, and impact as enumerated below.

1. FRET requires the molecules to be close together to give signal so typically molecules in solution are not used since they are far apart. The manuscript uses an uncommon method to determine the FRET parameters for these molecules and thus other peer reviewed papers should be referenced to support the method they use to perform FRET in solution. This would strengthen their argument that this is a valid method for FRET measurements.

2. Another curious statement on page 4, line 124, is that the authors were “satisfied with the qualitative result” for the transfer efficiency. The authors should justify why they do not wish to quantify the FRET efficiency.

3. The authors should include a paragraph at the end of the introduction to summarize the goals, hypotheses, and motivations for the work they did as these are not clear.

4. The method to determine solubility appeared to be non-standard. Typically, a shake flask method should be used to determine solubility by using a fixed time for sample to dissolve. Specific criteria were not given to determine when a sample was considered fully dissolved or not.

5. On page 6, line 167, the authors state that the same amount of triethylamine was added to each solution “to introduce the same amount of OH- ions to their system.” The only solvent that is protic is water. All others would not produce hydroxide ions. This phrase should be changed.

6. On page 6, line 169, 10-times the triethylamine concentration was used in THF because otherwise it “did not yield any effect.” What effect is not explained in the methods section here and it should be. Why not use this same concentration for the other solvents?

7. Line 191 page 6, pHema is not defined.

8. The preparation of samples in pHema and on paper fibers was referenced to their previous paper, ref 22. The authors should be clear in this paper the concentration of dye on fibers and in films and how this was determined.

9. The authors should justify with literature the calculation they used to determine the distance between dye molecules. A sample calculation should be provided.

10. The Table 1 data are not consistent with the description of the method. The method made is sound like all solutions were saturated, but the presented results suggest that only water was saturated. If this is the case, the method should be changed and the title of the table should be changed to concentration in solution as it is not the solubility limit. Also, the concentration of the dyes in water should be given in mmol/L to be consistent with the table results.

11. On line 239, the authors stated that they used the same amount of alkalinity by adding the same amounts of triethylamine (TEA) to the solvents. Since the solvents have different polarities, this statement is not true since the pKb of TEA will change. The discussion around the TEA addition is weakened by not knowing where the equilibrium of FTSC lies in terms of towards it absorbent or non-absorbent state.

12. The authors mention a NaOH reference on line 255, but do not state what this in reference to as it is not described in the methods.

13. The authors mention a chemical modification to the hydrazide group of the DCCH but they do not describe how this was done in the methods. For example, purification would affect the final pH of the mixture and the fluorescence behavior.

14. Since the method of pHema films with fluorescent dyes was not clear, it is not clear whether the concentration of the dyes are the same. This makes it difficult to assess the claim on page 9 line 292 that the QY increased as much as they state it did.

15. The QY measurement error seems high. What is the origin of the error? Is there an experimental method issue?

16. The sentence on line 436, “Due to this investigation… dyed with DCCH and FTSC.” Is confusing. The reference paper has already been published so the work of the current paper does not seem to have enabled that work. Also, all the current work was in solution not a pHema material.

17. In Table 4 what was the criteria for high or strong (low or weak) FRET signal? This analysis seems very qualitative.

18. The authors did not clearly discuss how solution based FRET can be translated to surface-surface interaction FRET.

6. PLOS authors have the option to publish the peer review history of their article (what does this mean?). If published, this will include your full peer review and any attached files.

Reviewer #1: Yes: SANJEEV R INAMDAR

Reviewer #2: No

---

## [Author Response · Author response to Decision Letter 0]

9 Jan 2020

As requested we responded in detail to the reviewers comments in the attached Response to Reviewers.

---

## [Editor Report · Decision Letter 1]

21 Jan 2020

Spectroscopic Investigation of DCCH and FTSC as a potential pair for Förster Resonance Energy Transfer in different solvents

PONE-D-19-23740R1

Dear Dr. Hirn,

We are pleased to inform you that your manuscript has been judged scientifically suitable for publication and will be formally accepted for publication once it complies with all outstanding technical requirements.

With kind regards,

Warren Batchelor

Academic Editor

PLOS ONE
---

## [Editor Report · Acceptance letter]

28 Jan 2020

PONE-D-19-23740R1 

Spectroscopic Investigation of DCCH and FTSC as a potential pair for Förster Resonance Energy Transfer in different solvents 

Dear Dr. Hirn:

I am pleased to inform you that your manuscript has been deemed suitable for publication in PLOS ONE. Congratulations! Your manuscript is now with our production department. 

With kind regards,

on behalf of

Dr Warren Batchelor 

Academic Editor

PLOS ONE